# Emulating Artificial Synaptic Plasticity Characteristics from SiO_2_-Based Conductive Bridge Memories with Pt Nanoparticles

**DOI:** 10.3390/mi12030306

**Published:** 2021-03-15

**Authors:** Panagiotis Bousoulas, Charalampos Papakonstantinopoulos, Stavros Kitsios, Konstantinos Moustakas, Georgios Ch. Sirakoulis, Dimitris Tsoukalas

**Affiliations:** 1Department of Applied Physics, National Technical University of Athens, Iroon Polytechniou 9 Zografou, 15780 Athens, Greece; babispapak@gmail.com (C.P.); stkitsio@mail.ntua.gr (S.K.); Konstantinos_mous@hotmail.com (K.M.); dtsouk@central.ntua.gr (D.T.); 2Department of Electrical and Computer Engineering, Democritus University of Thrace, 67100 Xanthi, Greece; gsirak@ee.duth.gr

**Keywords:** conducting filament, diffusivity, nanoparticles, synapses, plasticity, conductance, resistive memories

## Abstract

The quick growth of information technology has necessitated the need for developing novel electronic devices capable of performing novel neuromorphic computations with low power consumption and a high degree of accuracy. In order to achieve this goal, it is of vital importance to devise artificial neural networks with inherent capabilities of emulating various synaptic properties that play a key role in the learning procedures. Along these lines, we report here the direct impact of a dense layer of Pt nanoparticles that plays the role of the bottom electrode, on the manifestation of the bipolar switching effect within SiO_2_-based conductive bridge memories. Valuable insights regarding the influence of the thermal conductivity value of the bottom electrode on the conducting filament growth mechanism are provided through the application of a numerical model. The implementation of an intermediate switching transition slope during the SET transition permits the emulation of various artificial synaptic functionalities, such as short-term plasticity, including paired-pulsed facilitation and paired-pulse depression, long-term plasticity and four different types of spike-dependent plasticity. Our approach provides valuable insights toward the development of multifunctional synaptic elements that operate with low power consumption and exhibit biological-like behavior.

## 1. Introduction

The recent advances toward developing artificial neural networks and the perspective of carrying out in-memory calculations have increased the interest for emergent electronic synaptic elements that will be capable of performing multiple cognitive processes with low power consumption and high accuracy [1]. Even though several memory scenarios have been proposed toward that direction, a substantial amount of attention has been assembled around memristive devices, particularly due to their minuscule dimensions, relatively small power supply requirements and most important their excellent compatibility with the conventional complementary metal-oxide semiconductor (CMOS) procedures [2,3]. The comparative advantage of memristors, as far as the development of artificial synaptic devices is concerned, lies on their ability to continuously tune their synaptic weight values, which is quite crucial for the realization of various bio-synaptic properties [4]. 

In recent years, tremendous progress has been made toward emulating various neuromorphic functionalities with memristive devices [5]. The impressive characteristic of all the reported properties is that they cover a time constant between a few microseconds to several years. Under this light, we can argue that the synaptic properties are generally divided into two major categories: the short-term plasticity (STP) effects and the respective long-term plasticity (LTP) [6]. The STP pattern is associated with quick response and information filtering whereas LTP response is connected with the memory capacity and the dynamic modifications in synaptic strength. As concerns the STP properties, several types of plasticity effects that evolve within the range of microseconds to several minutes have been discovered, including the analog conductance modulation (excitatory/inhibitory responses), paired-pulsed facilitation (PPF), paired-pulse depression (PPD) and augmentation. In contrast with STP, LTP includes the long-term potentiation (LTPot), long-term depression (LTDep) effects as well as the transition from STP to LTP. The synaptic efficacy under LTP mode has a typical duration from several hours to years. Moreover, the distribution and firing activity of the incoming signals to the synapses from the neuron network, as well as the local dynamic synaptic weight modulation, results in the realization of some high-order synaptic plasticity characteristics with both long-term and short-term dynamics. These include the spike-dependent plasticity [7], the metaplasticity [8] and the hetero-synaptic plasticity effects [9]. Although the abovementioned list of synaptic properties seems daunting and it is almost impossible to devise an electronic element capable of performing all of these tasks, memristors arise as an ideal candidate for that direction. Despite their relatively simple metal–insulator–metal (MIM) structure, resistive switching devices possess extremely rich and complicated internal mechanisms capable of emulating a variety of bio-realistic synaptic properties [10].

Since the implementation of the various neuromorphic properties is strongly associated with the conducting filament (CF) underlying growth mechanism, we can attain reconfigurable synaptic responses by controlling the effective diameter of the CF. Consequently, we can argue that the STP functionalities can be emulated by small CF that exhibit unstable behavior whereas LTP behavior can be attained by bigger CFs that preserve their initial configuration even without applying pre-synaptic pulsing schemes. The presence of nanoparticles (NPs) within the proposed device structure can be proved quite beneficial toward controlling the potential percolation paths for the formation of the CFs, due to the intense local Joule heating effect and its influence on the thermal accelerated ion migration fluxes. As a result, we managed to reproduce several from the abovementioned synaptic plasticity effects, as well as four different types of spike-dependent plasticity, namely spike-rate dependent plasticity (*SRDP*), spike-voltage dependent plasticity (SVDP), spike-number dependent plasticity (SNDP) and spike-duration dependent plasticity (SDDP). A comprehensive numerical model was also applied in order to shed light on the CF dynamic behavior as well as the influence of the electrode thickness and material composition at the local Joule heating distribution. 

## 2. Materials and Methods

In order to assess the manifestation of various synaptic properties from our devices, we fabricated single layer devices of Ag/SiO_2_/TiN/Pt NPs. All thin film depositions were performed on oxidized silicon substrates by employing the RF magnetron sputtering technique. A high vacuum chamber with a base pressure of ~10^−6^ mbar was employed for the depositions that were carried out at room temperature. A high-purity ceramic target (SiO_2_ target = 99.99%) was used during the growth of the 20 nm thin film of SiO_2_. The flow of the inert gas (Ar) during the deposition was kept constant at 20 sccm while for O_2_ a lower flow was introduced into the vacuum chamber (1 sccm). The TiN bottom electrode (BE) and Ag top electrode (TE) were accordingly deposited from TiN and Ag sputtering targets with a total thickness of 15 and 40 nm, respectively. A total power of 140 W and 200 W was delivered onto the ceramic targets (SiO_2_, TiN) and metallic target (Ag), respectively. The fabricated devices were typical MIM capacitors with square electrodes of 100 um lateral dimension. The Pt NPs were deposited by using the novel gas condensation terminated technique from a high-purity Pt target (99.99% purity) [11]. This process evolves in a small chamber, attached through a small aperture (~5 mm) to the main one, where the samples are placed. The three main parameters that influence the size and density of the NCs are the DC current value that is used to ignite the plasma, the inert gas partial pressure within the condensation zone, which principally depends on the Ar flow, and the selected distance between the target material and the escaping aperture, which is defined as the length of the condensation zone. The latter can be adjusted from a minimum distance of ~10 cm (namely position 0) to a maximum of ~20 cm (namely position 100) and directly influences the size of the produced NCs. The bigger this distance, the larger the nanocrystal size achieved. For our case, a high Ar flow was chosen (60 sccm) in order to achieve a high surface density (~5 × 10^12^ NPs/cm^2^) of the deposited NCs (~5 nm average diameter). Moreover, a current value of 0.1 A was recorded during the deposition, which took place for about 20 min, while position 100 was imposed for the length of the condensation zone. 

The direct current (DC) I–V characteristics of the devices were performed with the Keithley 4200 semiconductor parameter analyzer (4200-SCS) at SUSS MicroTec probe station and at room temperature conditions. Retention measurements at elevated temperatures were carried out at a Janis Ltd. cryostat (−193 to 200 °C) probe station, under vacuum conditions (~10^−2^ mbar). The pulsed I–V measurements were conducted by employing the Keithley 4225 Pulse Measurement Unit (PMU). The structural properties of the Pt NPs were investigated by using the Philips CM 20 analytical High-Resolution Transmission Electron Microscope (TEM).

## 3. Results

### 3.1. DC I–V Characteristics

Figure 1a illustrates the DC I–V characteristics acquired after the application of external signals to the TE by keeping the BE to the ground mode. 

A compliance current (I_CC_) limit of 10^−4^ A was constantly applied in order to avoid the hard breakdown of the dielectric. It is also interesting to notice that no electroforming procedure was required prior to the memory device operation, which it is not the case for the majority of SiO_2_-based resistive switching memories [12]. Although the origins of this effect are not fully understood, we believe that the granular-like structure of the deposited film [13], which stems from the relatively high working pressures, leads to the creation of a nanoporous dielectric structure that facilitates the ionic migration [14]. The SiO_2_ thin film was found also in the amorphous state [15], which is also associated with the degree of porosity of a metal oxide thin film [16]. The sample operates under the bipolar switching mode and exhibits a quite big switching ratio (~10^6^), which is also beneficial for multibit storing capabilities [17]. The transition from the high-resistance state (HRS) to the low-resistance state (LRS) takes place at about V_SET_ ~ 220 mV while the opposite transition manifests at V_RESET_ ~ −50 mV. In addition, the sample exhibits a transition slope of ~10 mV/dec(A) (Appendix A) during SET transition. Interestingly, the sample without the presence of the 15 nm layer of TiN and only the employment of Pt NPs as BE presents a steeper transition slope (~1 mV/dec(A)—not shown here) where the reference sample with only 40 nm TiN as BE exhibits smoother transitions (~30 mV/dec(A)) [15]. It is thus apparent that not only the material selection (bulk metal or NPs) as well as the thickness of the BE decisively determine the switching pattern. Although the origins of this effect are still under investigation, it is conceivable that the surface roughness of the Pt NPs layer assist the creation of sharp edges that significantly enhance the local electric field [18]. Moreover, as can be ascertained from the TEM plan view image of the Pt NCs (inset of Figure 1a), the NCs do not exhibit any preferred morphology and several irregular shapes can be detected, which could likewise impact the electric field distribution. Moreover, the thermal conductivity values of the BE has a pronounced impact on the local Joule heat distribution [19], as is disclosed later in the analytical modeling section. The result of the abovementioned factors is the enhanced thermal accelerated migration of silver ions that could interpret the steeper transition slopes.

The cycle-to-cycle (temporal) and device-to-device (spatial) responses divulged no significant degradation of the hysteresis patterns after the application of 300 consecutive DC cycles (Figure 2a) or by measuring 100 different memory cells on the same sample (Figure 2b), respectively. In both cases, a large switching ratio is maintained (~10^5^), while as concerns the spatial distribution, low coefficients of variance (σ/μ) were measured for both the HRS and LRS, thus indicating the small dispersion of the respective I–V characteristics. 

### 3.2. Pulsed I–V Characteristics

The switching dynamics of our prototype was examined by enforcing square pulses of 100 ns width and ±0.5 V amplitude in order to impose the SET and RESET transitions, respectively. The results presented in Figure 3 indicate that a delay time (t_SET_) of ~80 ns is required in order to detect a discernible current increase and a delay time of t_RESET_ ~140 ns are required in order to reduce the operating current values. As far as the SET transition is concerned, smaller delay times (t_SET_) are clearly recorded as we increase the pulse amplitude from 0.5 to 2.5 V (Appendix A). More specifically, while a t_SET_ ~50 ns is required in order to switch the memory device into the LRS under the application of +0.8 V/100 ns square pulse, the same transition takes place at a t_SET_ ~20 ns when a higher pulse, in terms of amplitude is applied (+2.5 V/100 ns). In order to gain valuable insights into the origins of this effect, a series of measurements were carried out by enforcing various positive square pulses, with amplitude ranging from 0.5 to 2.5 V and step 100 mV [20]. The results depicted in Appendix A reveal a highly nonlinear connection between the pulse amplitude and the measured delay time, which indicates that the SET kinetics is governed by ionic hopping procedures. The latter mechanism exhibits a strong nonlinear relation from the local electric field and temperature distributions and assists toward the accelerated migration of silver ions through the silica matrix [21]. 

Pulse endurance and retention measurements were also performed on unstressed memory cells in order to assess their behavior after the application of a train of pulses (Figure 4). Square pulses of 1.5 V amplitude and 100 ns width were selected in order to attain an initial memory window of ~10^4^. Even though some variations are recorded after the enforcement of 10^8^ successive SET/RESET cycles, a switching ratio of 10^3^ is preserved, and this indicates the distinguishable nature of the two different resistance states. A comparable pattern was captured during the evaluation of the retention performance, where no remarkable fluctuations were measured for a period of 10^5^ s and at elevated temperatures (150 °C). 

### 3.3. Analytical Modeling

In order to elaborate on the underlying origins of the reported switching patterns as well as the CF growth mechanism, we used a comprehensive numerical model that was thoroughly presented in our previous work [15]. If we take into account that the switching effect within our conductive bridge memories (CBRAM) configuration originates from the anodic dissolution of metallic ions from the electrochemically active TE and their respective cathodic reduction at the BE [22], we can argue that a metallic chain composed of Ag nanoclusters (NCs) is progressively formed that is characterized as CF [23]. The filamentary conjecture is also in line with the independence of the resistance values of the LRS from the total device area, as can be ascertained from Appendix A. In our model, we have incorporated a term related to the thermo-diffusion flux of ions which is strongly affected by the local Joule heat distribution. Moreover, the whole device structure is taken into account within our theoretical approach, as well as the influence of the electrode material and thickness on the local temperature profiles. The simulated two-dimensional (2D) cross-section profile in is presented at Appendix A, where also the selected boundary conditions can be found. Truncated-cone CF was assumed during the numerical calculations in order to interpret the memristive responses during DC triggering [24]. Although this specific choice seems arbitrary, the simulation result reveals a quite good consistency with the respective experimental data. Moreover, there are in the literature tangible experimental pieces of evidence as concerns SiO_2_-based CBRAM that clearly indicate the existence of such types of CFs [25,26]. The whole memristive behavior can be modeled by calculating the evolution over time of the effective diameter (*φ*) distribution of the CF by employing the following three differential equations [27,28]:(1)$$\frac{d\varphi }{dt}={\left.\frac{d\varphi }{dt}\right\|}_{drift}+{\left.\frac{d\varphi }{dt}\right\|}_{diffusion}+{\left.\frac{d\varphi }{dt}\right\|}_{thermo-diffusion}=Ae^{-\frac{E_{drift}-aq\psi }{k_{B}T}}+B\varphi^{-1}e^{-\frac{E_{diff}}{k_{B}T}}-C\varphi^{-1}S\left(\frac{\partial T}{\partial r}+\frac{\partial T}{\partial z}\right)$$
(2)∇·σ∇ψ=0
(3)$$\rho_{m}C_{p}\frac{\partial T}{\partial t}=\nabla k_{th}\cdot \nabla T+\sigma \nabla {\left|\psi \right|}^{2}$$
where *α* is the barrier lowering factor, *E_drift_* is the energy barrier for ionic hopping, *E_diff_* is the diffusivity barrier, *ψ* is the electrical potential, *k_B_* is the Boltzmann constant, *T* is the absolute temperature and *S* is the Soret coefficient [29]. In Appendix A the values for all parameters used in our model, including the *A*, *B*, and *C* fitting constants, the electrical conductivity (*σ*), the mass density (*ρ_m_*), the thermal conductivity (*k_th_*), the activation energy of thermophoresis (*E_S_*) and the specific heat (*C_p_*) can be found. Moreover, the employed functions of thermal and electrical conductivity of the CF as a function of its effective diameter are presented at Appendix A. A numerical solver (COMSOL) was used in order to solve, simultaneously and self-consistently, Equations (1)–(3). The results that are presented at Figure 1a point out the validity of our theoretical approach while CFs with a diameter at the vicinity with the BE (*φ_B_*) of 6 nm was assumed (Appendix A). In Figure 1b, the 3D calculated maps of the effective diameter distribution of the CF are depicted for three different bias conditions. From these graphs, it is apparent that the switching effect originates from the modulation of the *φ_B_* near the BE since at this position the continuity of the CF starts to break. This result is anticipated since the 15 nm thick TiN layer exhibits a thermal conductivity value of about 1 Wm^−1^K^−1^ [30], while the 40 nm thick Ag layer possesses a significantly larger thermal conductivity value (~400 Wm^−1^K^−1^) [31]. It is thus obvious that the BE cannot efficiently dissipate the generated heat induced from the local current distribution. As a result, a large temperature gradient is formed that affects the stability of the CF. It is also interesting to notice that Ag also exhibits a greater thermal conductivity value from the commonly used electrode materials within valence change memories (VCMs) configuration (Pt, Au, Ti) [12]. As a result, the dissolution of the CF takes place normally near the TE/CF interface [32]. Since the TE surface is exposed to the environment, this effect could potentially lead to the device failure, due to the electrode delamination and the loss of the accumulated oxygen to the ambient [33]. 

### 3.4. Artificial Synaptic Activity

Within the brain configuration, the neurons encrypt information packages by emitting small voltage pulses that are called action potentials or spiking neuronal signals. In conjunction with the distribution of neurons within the brain structure, there is also a massive grid of synaptic elements that form a quite complex biological neural network, which is responsible for human perception, emotion, forgetting, learning and memory activities [34]. While neurons are responsible for the generation and propagation of action potential, the synapses execute the signal process and storage processes. It is thus obvious that the synapses rule the brain architecture and configure its structural plasticity and colossal parallelism for the assigned tasks. Within the brain, there are two types of synapses: the electrical synapses and the chemical synapses [35]. The pure electrical synapses represent the direct contact between the pre- and post-synaptic neurons and operate with relatively big frequency. However, they possess no flexibility and signal modulation properties and they are commonly encountered at the early fetus stages, where there is a lack of large-scale functional neural networks. In reality, the most ordinary type of synapses is the chemical synapses that exhibit a small gap (~20 to 50 nm) between the pre- and post-synaptic membranes. As a result, the synaptic efficiency or the junction strength can be adapted according to the activities of the interconnected neurons. This quite important effect is behind all the biological processes that take place within the human brain.

#### 3.4.1. Potentiation and Depression Responses

Figure 5 depicts the artificial PPF and PPD characteristics after the application of a train of total of 60 pulses with various amplitudes (Appendix A). During the execution of the PPF/PPD procedures, the second post-synaptic response becomes larger/smaller in respect to the first signal, through the application of two similar pre-synaptic stimuli. Our devices successfully reproduce these properties and also exhibit some other interesting features. While under the application of small pre-synaptic pulses of 100 mV amplitude a gradual modification of the post-synaptic current values is recorded for both PPF and PPD processes, an abrupt response is measured when increasing the pulse amplitude to 300 and 500 mV [36]. More specifically, after the application of 16 pre-synaptic signals with amplitude +500 mV and 43 signals with an amplitude of +300 mV, the post-synaptic response is quickly increased. A similar pattern takes place during the implementation of the PPD protocol, where after the application of 10 pre-synaptic pulses with amplitude −500 mV and 25 pulses of amplitude −300 mV, the current values are abruptly reduced. These traits are in line with the respective responses acquired during the application of DC signals since the SET transition takes place always at a higher bias with respect to RESET transition. In addition, the higher the triggering amplitude the smaller pulsing sequence is required in order to impose the targeted conductance state. The above effects can be viewed as direct evidence during the implementation of bio-synaptic properties [37]. 

#### 3.4.2. STP and LTP Effects

In order to leverage from the abovementioned transient responses, we thoroughly investigated the memory loss properties of our devices. According to the well-known model of Atkinson and Shiffrin, the human memory is composed of three different components: the sensory part and the short-term and long-term memories [38]. The sensory memory stores temporarily the incoming information from the vast network of the sensory systems that possess the human body (i.e., tactile, auditory, visual, etc.). If the information packages are received at a higher rate, sensory memory is transformed to short-term memory. Accordingly, a transition from short-term to long-term memory can be imposed by controlling the properties of the pre-synaptic signals, such as the amplitude and the frequency [39]. In order to monitor the transformation from STP to LTP, as far as the potentiation process is concerned, we applied a train of pulses with different frequencies. As can be ascertained from Figure 6a, our sample retains the current values of the post synaptic response even in the case of reducing the frequency of the pre-synaptic signals from 1 MHz to 100 kHz. The progressive growth of the measured current under the application of 1 MHz signals is attributed to the enlargement of the effective diameter of the CF, as it is schematically depicted in Figure 6c–e. Further experiments were also carried out by enforcing a different number of pre-synaptic pulses (20, 40 and 60) with high frequency (1 MHz) and constant amplitude of 500 mV and monitoring the current decay rate through the application of 95 pulses with a frequency of 100 kHz and same amplitude [40]. The results are presented at Figure 6b, where the normalized current decay distribution is shown. An exponential relaxation approach is followed in order to describe the memory loss properties [41]:(4)$$I_{t}=I_{0}+A_{I}e^{\left(-\frac{t}{\tau }\right)}$$
where *I_t_* is the current value after the elapse of time *t*, *I_0_* is the initial current value, *A_I_* is a fitting constant and *τ* is the relaxation time. From the fitting results, we can draw the conclusion that the number of successive stimulations has a direct impact on learning efficiency. The extracted relaxation times suggest the successful emulation of the LTP effect since a long-lasting memory performance is achieved and is enhanced as we increase the number of the pre-synaptic signals [42]. From a physical point of view, we can argue that the formation of relatively big and robust CF that cannot self-rupture, in terms of effective diameter distribution, is the driving force for this effect [43].

However, a different picture takes place when we enforce smaller pre-synaptic signals, in terms of amplitude. Although initially a gradual increase of the measured current values is recorded, during the application of high-frequency pre-synaptic signals (1 MHz), by reducing the frequency the current responses quickly decay (Figure 7a). This behavior is comparable with the memory forgetting pattern of the human brain and signifies the manifestation of the STP effect [44]. The respective relaxation times acquired through the fitting process are also compatible with this assumption (Figure 7b). It is also interesting to notice that the current values of the post-synaptic signals are not completely diminished to the initial off-state, fact that indicates the existence of an attenuated LTP. This behavior is attributed to the existence of small CFs with sub-nm scale diameters that exhibit unstable nature (Figure 7c–e). In any case, our prototypes can also reproduce the rich dynamics of the synaptic plasticity that exists within the biological brains and comprises temporary and more stable constituents (i.e., the STP and LTP) [45]. Similar patterns are observed for the emulation of short-term depression (STD) and LTD (not shown here). 

#### 3.4.3. Synaptic Weight Modulation

Figure 8 highlights the artificial synaptic excitatory and inhibitory responses acquired after the application of a train of 120 square pulses with 100 ns width and 1 V amplitude. These pulse characteristics were chosen in order to detect discernible patterns on the extracted conductance values. The data were assembled from 20 different memory elements, on the same sample, whereas the recorded variations could be related to the random formation of the CF and the diameter-related uniformity issues of the Pt NPs [46]. The conductance modulation pattern takes place in two stages. Initially, a sharp increase/decrease of the respective conductance values takes place after the application of 12 potentiation/9 depression pre-synaptic pulses. Afterward, a plateau is recorded for both synaptic processes. This behavior is closely associated with the steep transitions during that the hysteresis pattern presents and it is detrimental in terms of efficiency of the backpropagation algorithm at the training stage of deep neural networks (DNNs) [47]. Several optimization procedures have been proposed toward obtaining a more linear response, such as the incorporation of bilayer structures [37], but at the expense of the capability of emulating the STP to LTP transition [48,49]. 

#### 3.4.4. Spike-Dependent Plasticity Effects

Taken into account that spiking neural networks (SNNs) are more promising in terms of low power consumption and enhanced capabilities of processing real-time patterns [50], we examined the manifestation of various spike-dependent plasticity effects from our devices. Firstly, we investigated the implementation of the *SRDP* effect by enforcing a pair of square pulses with the same amplitude and width but different delay time Δt (from 100 to 900 ns) [51]. The results are presented at Figure 9a,b, while more synaptic profiles can be found at Appendix A. The distribution of the *SRDP* index, which is presented at Figure 9c, can be well fitted by applying the following expression [52]:(5)$$SRDP=C_{1}e^{\left(-\frac{t}{\tau_{1}}\right)}+C_{{}_{2}}e^{\left(-\frac{t}{\tau_{2}}\right)}$$
where *C*_1_ and *C*_2_ are fitting constants and *τ*_1_ and *τ*_2_ are the respective time scales that describe the dynamics of the whole effect. The drop of the *SRDP* index is strongly correlated with the structural properties of the CF. From our analysis, we have calculated the values of 90 and 110 ns for the *τ*_1_ and *τ*_2_ times, respectively, which are in line with the relaxation times of the CF that were reported in our previous work [15]. This outcome indicates that the state stability of the CF does not only influences the memory-related properties but also provides unique opportunities toward emulating a broad range of biological synaptic properties since the *SRDP* can be regarded as an extended form of the Hebbian-type spike-time-dependent plasticity, which is associated with the learning and forgetting tasks of the memory [53].

Three other types of spike-dependent plasticity were investigated, including SVDP by applying a set of square pulses with increasing amplitude (from 0.5 to 2.5 V), SNDP by enforcing an array of square pulses with different pulse number (from 1 to 54) and SDDP by implementing a set of square pulses with enlarging width (from 100 to 500 ns). The results are presented in Figure 10, and more measurements can be found in Appendix A. It is interesting to notice that all indexes related to these three biological effects exhibit a discernible growth that is related to the respective learning procedures of the STP through consecutive rehearsal routines [54]. On the other hand, the *SRDP* index is clearly reducing, suggesting that it is linked with the forgetting tasks of the STM. These results can be interpreted in terms of modulating the effective diameter of the CF and are of great importance for the realization of memristive neural networks with inherent neuromorphic properties.

## 4. Discussion

From the above-reported results, it is apparent that the dynamics of the CF formation and annihilation processes dictate the whole memristive pattern. A direct comparison with reference samples can be found in our previous works, in terms of resistive and synaptic performance properties. The comparative advantage of the NPs’ incorporation is the ability to emulate both STP and LTP artificial synaptic responses in one single switching mode [55,56]. Similar results have been also reported for other material configurations [57,58,59]. Moreover, we have to underline that Pt NPs facilitate the charge transfer from the BE to SiO_2_, since a lower interface barrier is formed [60], in terms of height, which increases the transmission probability for electron tunneling. This assumption is also in line with the experimental data of the retention performance (Figure 4b), thus highlighting the acceptable behavior of our prototype. Additionally, in our previous work, we have dealt with similar issues, regarding the charge transfer properties of Pt and Ta NCs within the TiO_2_ dielectric matrix [61].

Although the impact of Ag active TE on SiO_2_-CBRAMs has been studied by other groups [62,63], the main contribution of the present work is the demonstration of a relatively sharp transition slope, low V_SET_ voltage and huge memory window. Moreover, we have to underline the completely forming-free nature of the proposed memory concept that is always beneficial in terms of periphery circuit design. Our results are also in line with first-principle calculations from the literature [64], where the formation of silver ions is directly associated with the oxygen content of the metal oxide film. This result could interpret the forming free nature of our full stoichiometric memory devices.

The thermal runaway effect induced from the local Joule heat distribution as well as the ionic displacement rate is regarded as two crucial factors that define the kinetics of the resistive switching effect [65,66]. All of these effects are incorporated in our model that relies on the size-dependent melting procedure of Ag NCs that are produced due to material precipitation as a result of the enhanced solid-solubility of Ag [67]. An interesting effect arises here, since the melting point of metallic NCs presents a strong dependence on their diameter [68]. As a result, the melting temperature is quickly falling, while the diameter of the NCs is getting smaller. This phenomenon has a direct impact on the evolution of the switching effect, as well as its volatile or non-volatile nature [15]. More specifically, if the local temperature profiles exceed the meting point of the formed Ag NCs, a transition from threshold to bipolar switching can take place. The melting point of 6 nm Ag NCs has been estimated at around 700 K [69,70], while, during set transition, lower temperature values are developed within our device active core (Figure 11a). For that reason, a negative bias is required in order to break the CF continuity and create a tunneling gap, since the imposed local temperature distribution now exceeds the melting point of the Ag NCs (Figure 11b). Interestingly, by employing a thicker layer of TiN (40 nm) as BE the simulation outcome predicts even smaller temperature values during the SET transition due to the increased thermal conductivity value of the thicker TiN film (Appendix A). On the other hand, the incorporation of only the layer of 5 nm thick Pt NPs yields in relatively high local temperature distributions that can melt the respective Ag NCs even at a positive bias (Appendix A). This effect originates from the small thermal conductivity value of Pt NPs, which is one order of magnitude smaller than 15 nm TiN and two orders of magnitude smaller than the 40 nm TiN [71,72]. Similar effects have been reported in the literature regarding the impact of the BE thickness on the manifestation of various switching modes [73].

By combining the layers of Pt NPs and 15 nm TiN, we limit the generated heat due to the relatively bigger thermal conductivity value of the latter film, and, at the same time, we leverage from the surface roughness of the Pt NPs toward the local electric field enhancement. Moreover, the increased local temperature accelerates the ionic migration rate. The common denominator of all of these effects is the manifestation of a bipolar switching mode with intermediate transition slopes that is beneficial toward emulating not only the STP and LTP biological responses but several types of spike-dependent plasticity characteristics.

## 5. Conclusions

In conclusion, we demonstrated that a broad range of artificial synaptic property characteristics can be emulated by SiO_2_-based memristors, with Pt NPs as BE, after the application of low-power pulses. The intermediate transition slope permits the manifestation of both STP and LTP effects, while four different types of spike-dependent plasticity were successfully reproduced by our device. Fruitful insights regarding the influence of the BE configuration on the whole memristive pattern by the application of a self-consistent numerical model are presented, including the key role of the ionic diffusivity and the thermal conductivity of the operating electrodes. Our results indicate that, with proper material engineering, it is feasible to emulate a wide range of neuromorphic properties that are quite important for the hardware-based realization of memristive neural networks that will operate with the spike-time dependent plasticity algorithm.

## Figures and Tables

**Figure 1 micromachines-12-00306-f001:**
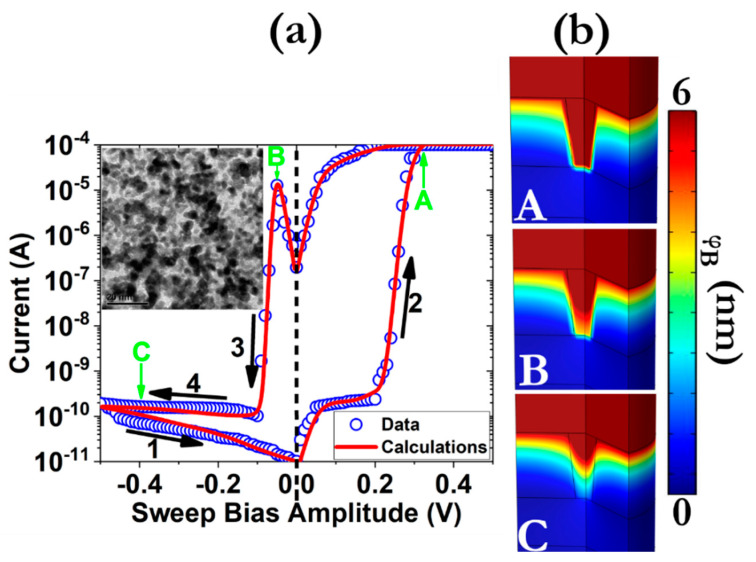
(**a**) Measured and calculated I–V hysteresis patterns during the application of 0.5 V external bias, under the application of a compliance current (I_CC_) limit of 100 μA. The inset picture depicts a TEM plan view image of the dense layer of Pt of nanoparticles (NPs). The scale bar in the TEM figure corresponds to 20 nm. The arrows in the graphs indicate the switching direction, while the sweep rate was 10 mVs^−1^. Similar switching patterns were captured by beginning the sweeps from 0 V to |Vmax| (not shown here). (**b**) 3D calculated maps of the conducting filament (CF) effective diameter distribution for states A (+0.3 V), B (−0.05 V) and C (−0.4 V).

**Figure 2 micromachines-12-00306-f002:**
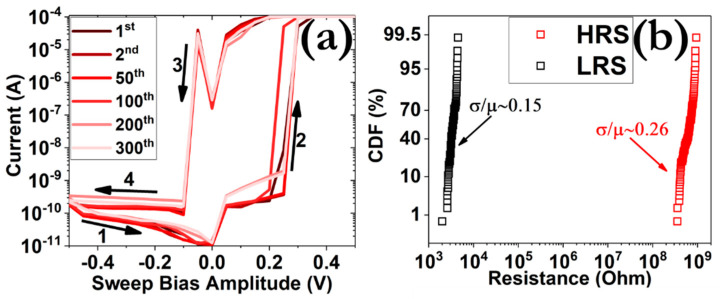
(**a**) I–V cycling behavior during the application 300 consecutive hysteresis loops and (**b**) cumulative distribution functions (CDF) of both low-resistance state (LRS) and high-resistance state (HRS) responses during DC consecutive operation from 100 different memory cells on the same sample. The resistance values were extracted at a read out voltage of 100 mV.

**Figure 3 micromachines-12-00306-f003:**
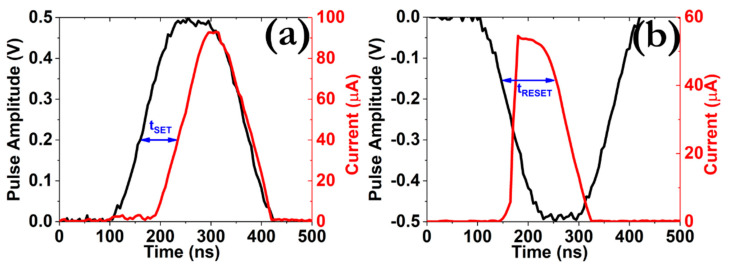
Measured transient current responses (red line) as a function of time during the application of (**a**) + 0.5 V/100 ns and (**b**) −0.5 V/100 ns square pulses (black line). A limit of I_CC_ = 1 mA was imposed in all cases.

**Figure 4 micromachines-12-00306-f004:**
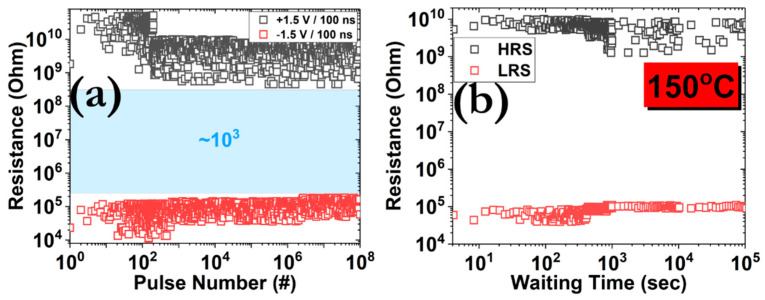
(**a**) Pulse endurance measurements under the application of ± 1.5 V/100 ns and (**b**) retention measurements at 150 °C (read voltage pulse 100 mV/100 ms).

**Figure 5 micromachines-12-00306-f005:**
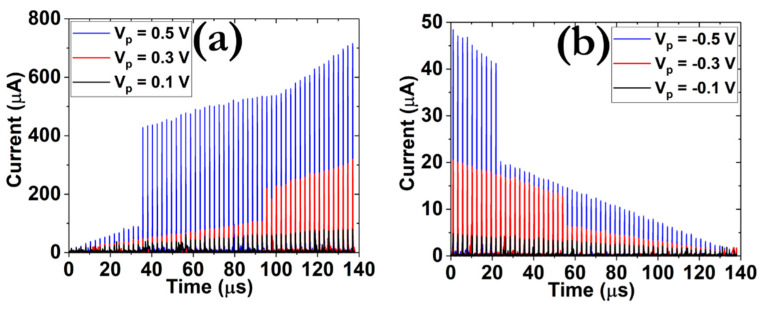
Distribution of the post-synaptic current responses during the implementation of the (**a**) potentiation and (**b**) depression procedures, under the application of total 60 pre-synaptic pulses with 100 ns width and amplitudes of ±0.1, ±0.3 and ±0.5 V, respectively.

**Figure 6 micromachines-12-00306-f006:**
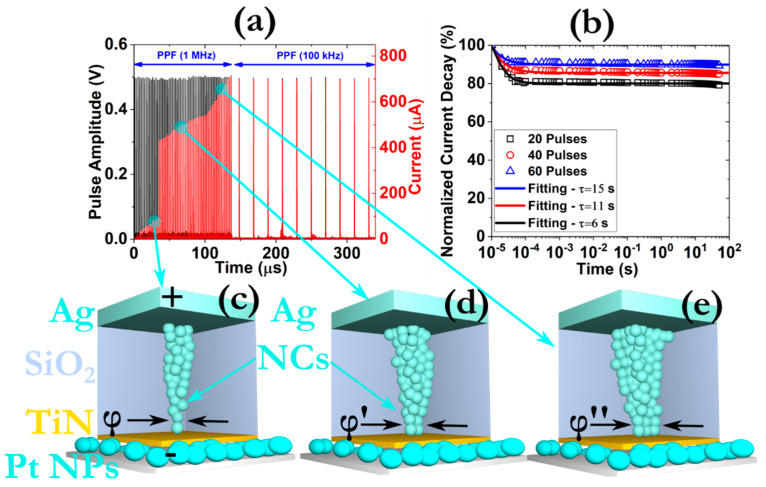
(**a**) Experimental verification of the paired-pulsed facilitation (PPF) effect by monitoring the post-synaptic current response after the application of a train of 70 pulses with the same width (100 ns), amplitude (0.5 V) and different frequency; (**b**) normalized current decay distribution as a function of time demonstrating long-term plasticity (LTP) synaptic behavior. The data were collected after applying a set of 95 pulses, with a frequency of 100 kHz, after the enforcement of 60 pulses, with frequency 1 MHz; (**c**–**e**) schematic representation of the CF growth procedure during the application of the consecutive pulsing scheme. The symbol *φ* stands for the effective diameter of the CF.

**Figure 7 micromachines-12-00306-f007:**
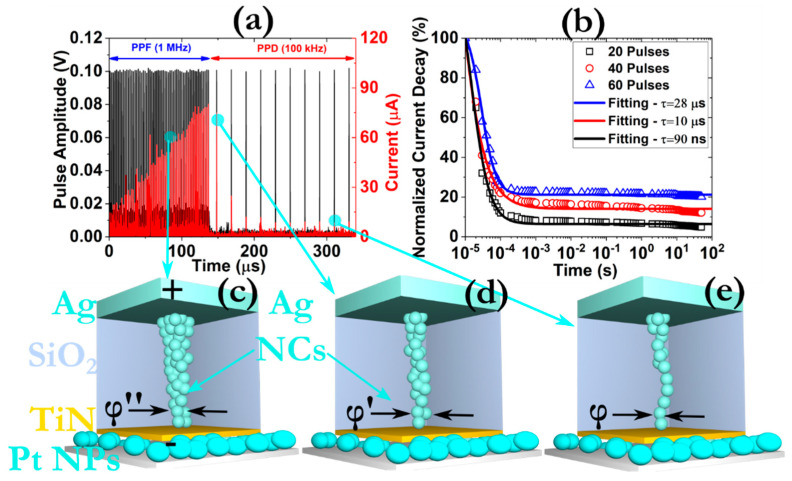
(**a**) Experimental verification of the PPF to paired-pulse depression (PPD) transition effect by monitoring the post-synaptic current response after the application of a train of 70 pulses with the same width (100 ns), amplitude (0.5 V) and different frequency; (**b**) normalized current decay distribution as a function of time demonstrating STP synaptic behavior. The data were collected after applying a set of 95 pulses, with a frequency of 100 kHz, after the enforcement of 60 pulses, with frequency 1 MHz; (**c**–**e**) schematic representation of the CF growth procedure during the application of the consecutive pulsing scheme. The symbol *φ* stands for the effective diameter of the CF.

**Figure 8 micromachines-12-00306-f008:**
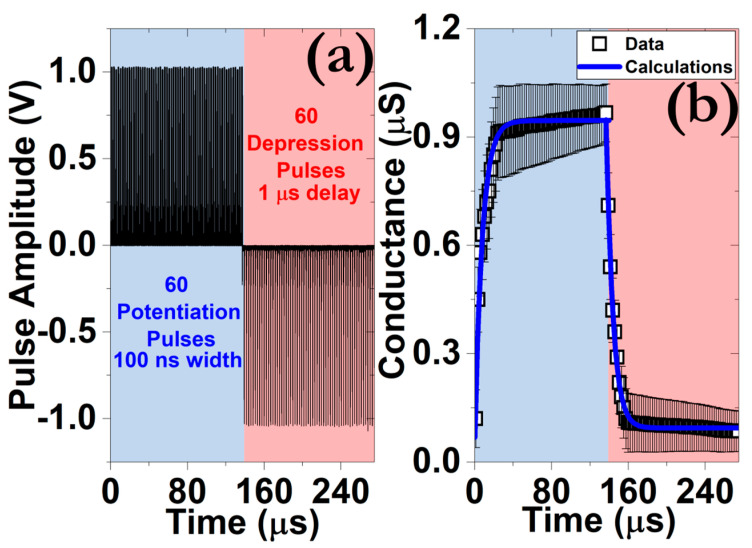
(**a**) Profile of the pre-synaptic voltage signals during the application of a train of 120 pulses with ±1 V amplitude, 100 ns width and 1 μs delay time. The sign of the pulse sequence was inverted after the application of 60 identical pulses. (**b**) Continuous modulation of the post-synaptic conductance during the potentiation and depression procedures. The data were collected by using a read voltage pulse of 100 mV/100 ms.

**Figure 9 micromachines-12-00306-f009:**
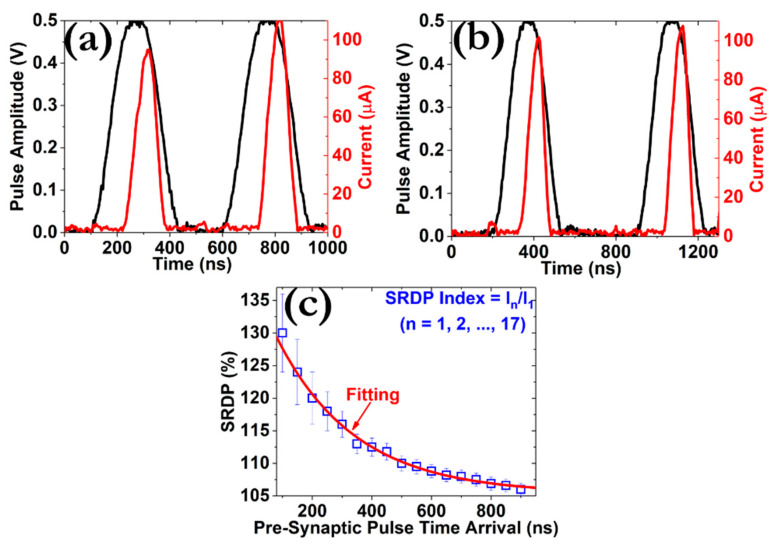
Demonstration of the PPF effect as a function of the time interval between the pre-synaptic pulses. Seventeen different pulse intervals were tested by applying two successive pulses of 0.5 V/100 ns; (**a**,**b**) depict the post-synaptic responses for a delay time of 100 and 200 ns, respectively. (**c**) Distribution of the spike-rate dependent plasticity (*SRDP*) index (I_n_/I_1_) calculated by two consecutive pulses with time interval 100 ≤ Δt ≤ 900 ns (I_1_, 1st post-synaptic current; I_n_, n^th^ post-synaptic current, n = 1, 2, 3, …, 17).

**Figure 10 micromachines-12-00306-f010:**
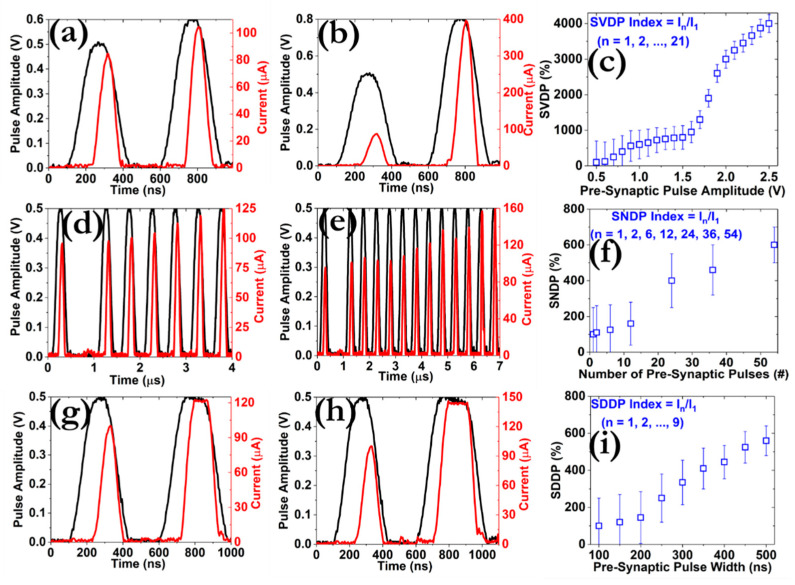
Demonstration of the PPF effect as a function of the amplitude between the pre-synaptic pulses. Twenty-one different pulse amplitudes were tested by applying two successive pulses of 100 ns width and delay time, while (**a**,**b**) depict the post-synaptic responses for pulse amplitude of 0.6 V and 0.8 V ns, respectively. The amplitude of the first pre-synaptic pulse was always kept constant at 0.5 V. (**c**) Distribution of the spike-voltage dependent plasticity (SVDP) index (I_n_/I_1_) calculated by two consecutive pulses with amplitude 0.5 ≤ V_p_ ≤ 2.5 V (I_1_, 1st post-synaptic current; I_n_, nth post-synaptic current, n = 1, 2, 3, …, 21). Demonstration of the PPF effect as a function of the number of pre-synaptic pulses. Seven different pulse sequences were tested by applying pulses of 0.5 V amplitude, 100 ns width and delay time, while (**d**,**e**) depict the post-synaptic responses by enforcing a total number of the pulse train of 6 and 12, respectively. The train of the pulses was enforced after the application of a single pulse of 0.5 V amplitude and 100 ns width, while the delay time was set to 1 μs. (**f**) Distribution of the spike-number dependent plasticity (SNDP) index (I_n_/I_1_) calculated by the number of spikes (n = 1, 2, 6, 12, 24, 36, 54). Demonstration of the PPF effect as a function of the width of two consecutive pre-synaptic pulses. Nine different pulse durations were tested by applying pulses of 0.5 V amplitude and 100 ns delay time, while (**g**,**h**) depict the post-synaptic responses for a pulse width of 150 and 200 ns, respectively. The width of the first pulse was always kept constant at 100 ns. (**i**) Distribution of the spike-duration dependent plasticity (SDDP) index (I_n_/I_1_) calculated by the number of spikes (n = 1, 2, 3, …, 9).

**Figure 11 micromachines-12-00306-f011:**
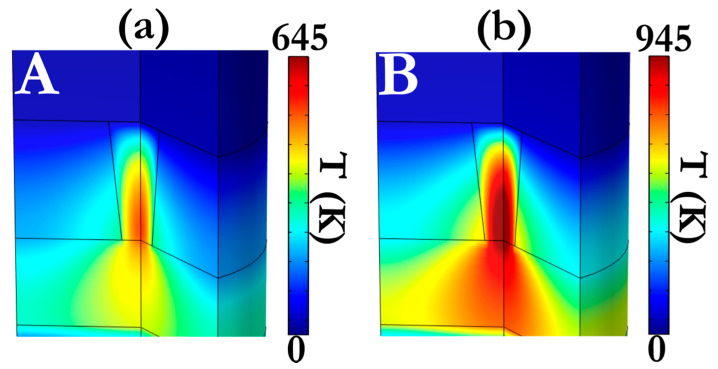
Three-dimensional calculated maps of the localized temperature distribution for (**a**) state A (+0.3 V) and (**b**) state B (−0.05 V).

## Data Availability

The data that support the findings of this study are available from the corresponding author upon reasonable request.

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
