# Peer review of "Emulating Artificial Synaptic Plasticity Characteristics from SiO2-Based Conductive Bridge Memories with Pt Nanoparticles"

_micromachines, 2021, doi:10.3390/mi12030306_

Round 1

Reviewer 1 Report

This paper studies the artificial synaptic plasticity characteristics from SiO2-based bridge memories with Pt nanoparticles. The authors performed detailed analysis on the interfacial and charge transfer effects of the system for real applications. This is an interesting study, with the topics well suits for the journal. The authors performed a number of important analysis and characterizations which shows a clear structure-performance relationship. The results are of good significance and novelty. The manuscript was well-written and properly organized. Before it is acceptable, I only have several minor concerns that should be carefully addressed by the authors.

Therefore, I recommend that this nice piece of work is acceptable for the journal after a minor revision. Details of my comments are shown as follow.

  1. The potential long-range charge transfer from Pt to the semiconductors like SiO2 can be discussed.
  2. It would be interesting to compare the results with previous experiments and/or first principle calculations from literature.
  3. There are several minor typos in the manuscript. Please double-check the paper.
  4. It would be discuss other 4d and 5d transition metals such as Pd and Au, in addition to Pt.
  5. It would be better to also discuss the effects if using ZrO2/HfO2 or other semiconductors/insulators instead of SiO2.

Reviewer 2 Report

Bousoulas P. et al. discussed the synaptic behavior of SiO2-based CBRAM for neuromorphic computing applications; the electrical characteristics have been studied very well and this work could benefit its field. However, before I can recommend the publication of this manuscript, the authors should address several problems below:

1. The manuscript is titled “…with Pt nanoparticle”, but this work does not discuss the devices made without Pt nanoparticles (nor with bare Pt film ) as a fair comparison or reference/control sample.

2. Fig.2(a) and Fig. 5 do not exactly indicate good analog behavior like typical artificial synapses do -since the current rises abruptly instead of gradually; In fact, the potentiation and depression data in Fig 8(b) shows a typical digital behavior. The authors should provide justification why they think their device could work for real AI hardware that requires a highly linear synaptic.

3. Literature study regarding artificial synapse devices is not adequate. The authors should study and cite these papers to concise the manuscript; Neutral oxygen irradiation enhanced forming-less ZnO-based transparent analog memristor devices for neuromorphic computing applications (DOI:10.1088/1361-6528/ab7fcf), Enhanced Synaptic Linearity in ZnO-Based Invisible Memristive Synapse by Introducing Double Pulsing Scheme (DOI: 10.1109/TED.2019.2941764), ZnO2/ZnO bilayer switching film for making fully transparent analog memristor devices (DOI:10.1063/1.5092991)

Minor:

  1. The acronyms should be first mentioned in the introduction instead of in the abstract, unless the acronym used more than once in the abstract.
  2. I suggest long-term depression uses LTDep acronym, for consistency.
  3. Please add label (a) and (b) in Fig 8.

Round 2

Reviewer 2 Report

The authors have addressed my comments and corrected the issues accordingly. I recommend the publication of this manuscript.